# Preventive Health Behavior and Readiness for Self-Management in a Multilingual Adult Population: A Representative Study from Northern Italy

**DOI:** 10.3390/nursrep15070240

**Published:** 2025-07-01

**Authors:** Dietmar Ausserhofer, Christian J. Wiedermann, Verena Barbieri, Stefano Lombardo, Timon Gärtner, Klaus Eisendle, Giuliano Piccoliori, Adolf Engl

**Affiliations:** 1Institute of General Practice and Public Health, Claudiana—College of Health Professions, 39100 Bolzano, Italy; 2Claudiana Research, Claudiana—College of Health Professions, 39100 Bolzano, Italy; 3Provincial Institute for Statistics of the Autonomous Province of Bolzano—South Tyrol (ASTAT), 39100 Bolzano, Italy; 4Directorate, Claudiana—College of Health Professions, 39100 Bolzano, Italy

**Keywords:** preventive health behavior, patient activation, health literacy, multilingual population, mistrust in healthcare, GHP-16, South Tyrol, public health promotion

## Abstract

**Background/Objectives:** Preventive health behaviors are key to disease prevention and health system sustainability; however, population-level factors remain understudied in multilingual regions. South Tyrol, an autonomous multilingual province in Northern Italy, provides a unique setting to examine how sociodemographic and linguistic factors shape preventive behaviors. **Methods:** A stratified, population-representative survey of 2090 adults (aged ≥18 years) was conducted in South Tyrol in 2024. Preventive behavior was assessed using the validated 16-item Good Health Practices Scale (GHP-16). Key predictors included age, sex, education, living situation, language group, employment in the health/social sector, health literacy (HLS-EU-Q16), patient activation (PAM-10), and the mistrust of professional health information. Weighted statistics and multivariable linear regression were used to identify associations. **Results:** The GHP-16 scores varied significantly across sociodemographic and psychosocial strata. Female sex, older age, higher education, higher patient activation, and sufficient health literacy were independently associated with greater engagement in preventive behaviors. Patient activation showed the strongest graded effect (β = 1.739). The mistrust of health professionals was inversely associated with behavior (β = –0.050, 95% CI: –0.090 to –0.009). Italian speakers reported higher GHP-16 scores than German speakers (β = 0.377), even after adjusting for covariates. Item-level analysis revealed small but consistent differences, particularly in information-seeking and vaccination behavior. **Conclusions:** Preventive behaviors in South Tyrol are shaped by demographic, psychosocial, and linguistic factors. Patient activation and health literacy are key modifiable predictors, whereas language group differences suggest structural and trust-related disparities that require tailored public health strategies in multilingual settings.

## 1. Introduction

Health-promoting behavior is a cornerstone of chronic disease prevention and long-term well-being, particularly in the aging population. The Good Health Practices (GHP-16) scale [1], a condensed and validated subset of the original Health Behavior Checklist (HBC) [2], provides a standardized measure of individual engagement in preventive health practices, such as regular exercise, adequate sleep, preventive medical care, and non-smoking.

Studies using the HBC or GHP-16 have identified key individual differences that predict variations in health behavior [1]. Notably, health literacy and self-efficacy have been shown to predict engagement in wellness maintenance and lower involvement in substance use behavior. In a study of college students with chronic illnesses, both higher health literacy and self-efficacy were associated with more frequent general health behaviors and wellness maintenance [3]. Moreover, personality factors such as conscientiousness and perceived health competence have emerged as robust predictors of preventive behavior in older adults and are linked to better exercise, dietary, and information-seeking behaviors, even after controlling for health status [4].

Cross-national and regional comparisons underscore the phenomenon that culturally distinct populations exhibit different patterns in health practices, validating the need for locally representative surveys. For instance, a Czech national study mapped age-related changes in health-related behavior and found significant cohort effects across dietary, physical activity, and substance avoidance dimensions [5].

South Tyrol, an autonomous province in Northern Italy, presents a unique setting for studying population health behavior. The region is characterized by its bilingual population (German- and Italian-speaking groups), longevity exceeding national averages [6], and distinct sociopolitical features, including low institutional trust [7], frequent reliance on complementary and alternative medicine [8], and notably low vaccine uptake [9]. Previous research has identified substantial differences in health literacy and health information use between German- and Italian-speaking populations in South Tyrol [10]; however, it remains unclear to what extent these disparities translate into differences in everyday preventive health behaviors in the general population. This study is the first to examine preventive health behavior using the GHP-16 scale, investigating its relationships with mistrust, health literacy, and patient activation in a representative multilingual population, and providing a framework to understand self-management readiness in the region’s diverse healthcare context.

Building on a representative, population-weighted survey of adults in South Tyrol, this study describes engagement in health-promoting behavior using the GHP-16 scale. We examined behavioral variations across sociodemographic subgroups and assessed whether mistrust in professional health information, health literacy, and patient activation independently predicted preventive practices. The following research questions guide this analysis:How does preventive health behavior, as measured by the GHP-16 scale, vary across sociodemographic subgroups (e.g., age, sex, education level, and linguistic group) in South Tyrol?Is mistrust in health information from professional sources associated with lower GHP-16 scores, independent of sociodemographic and psychosocial factors?To what extent are health literacy and patient activation associated with GHP-16 scores after adjusting for sociodemographic variables?

The results of this study can be interpreted using the Health Belief Model (HBM), which conceptualizes preventive health behavior as influenced by the perceived susceptibility and severity of illness, perceived benefits and barriers to action, cues to action, and self-efficacy [11]. In this context, health literacy and patient activation likely reflect enabling constructs, such as perceived control and confidence in managing health, while mistrust in professional health information may act as a barrier to engaging in recommended behaviors. Language group affiliation may serve as a culturally embedded cue to action, shaping exposure to and trust in health messages. Figure 1 illustrates how this study’s variables align with the core domains of the HBM, offering a framework for understanding the observed behavioral patterns in a socio-culturally diverse population.

By addressing these questions, this study contributes to a better understanding of population-level behavioral patterns in decentralized health systems. The results may support public health strategies that are responsive to social and cultural heterogeneity and help identify groups that may benefit from strengthened health communication and behavioral support. Given the increasing role of nurses in delivering preventive care and health education, particularly in culturally diverse settings, understanding how psychosocial and linguistic factors influence preventive behaviors is directly relevant to community-based and public health nursing practice.

## 2. Methods

### 2.1. Study Context and Sampling Framework

This study draws on a representative cross-sectional survey conducted between March and May 2024 in the Autonomous Province of Bolzano (South Tyrol), Italy. The province is a linguistically and culturally diverse alpine region with distinct population health patterns and a decentralized health system. The survey was jointly developed and administered by the South Tyrolean Institute of Statistics (ASTAT) and the Institute of General Practice and Public Health. This analysis is part of a series of secondary analyses derived from a comprehensive population-representative health survey conducted in South Tyrol in 2024. Previous publications based on this dataset have addressed sleep quality [12] and health information use and trust [10], using distinct outcome measures and focusing on different research questions.

A stratified random sampling design was employed to ensure broad population representation. The sampling strata included key sociodemographic dimensions: age category (18–34, 35–54, and ≥55 years), biological sex, nationality, and municipality of residence. A total of 4000 individuals were randomly selected from the provincial population registry to provide robust estimates across strata and enable post-stratification weighting using the ReGenesees software (version 2.3, Rome, Italy). The survey targeted all adults aged 18 years and older who resided in South Tyrol. Respondents completed the survey in either German or Italian using a standardized online or paper-based format. Participation was voluntary, and informed consent was obtained prior to the data collection.

### 2.2. Sociodemographic and Health Characteristics

Respondents provided detailed sociodemographic information, including age (in years), gender (male, female), linguistic group (German, Italian, Ladin, or other), and citizenship (Italian or other). Education level was classified into four categories: primary, vocational/technical, secondary, and university degree. Urbanicity was coded based on municipality type, and cohabitation status was recorded as living alone or with others (e.g., family or partner).

Self-perceived health was rated on a continuous visual analog scale ranging from 0 (worst imaginable health) to 100 (best imaginable health), and self-rated health was grouped into three categories: “high” (>80), “moderate” (60–80), and “low” (<60). Chronic illness was assessed using a self-report checklist, indicating whether the participant had been diagnosed with one or more chronic health conditions. For the analysis, both a binary indicator (presence vs. absence of any chronic condition) and a count variable (number of conditions) were created.

### 2.3. Assessment of Health Behavior (GHP-16)

Health-promoting behavior was assessed using the GHP-16 scale, a validated short version of the original 40-item HBC proposed by Hampson et al. The GHP-16 captures a broad range of positive health behaviors, including physical activity, diet, preventive care utilization, oral hygiene, vaccination, and avoidance of harmful substances. Each item in our survey was rated on a 4-point Likert scale ranging from 1 (“does not apply at all”) to 4 (“fully applies”). To enable meaningful comparison with international studies that used a 5-point Likert scale (ranging from 1 to 5), all GHP-16 item responses, originally collected on a 4-point scale (1 = “does not apply at all” to 4 = “fully applies”), were linearly rescaled to the 1–5 metric. This transformation was applied using the following formula: rescaled score = 1 + (original score − 1) × (4/3). This approach preserved the ordinal structure, spacing, and relative differences between response options while aligning the scale with the original GHP-16 format. Higher scores after transformation continued to reflect a greater frequency or intensity of health-promoting behavior, ensuring compatibility with the scoring conventions used in international comparative studies. For descriptive comparison and effect size analyses, the rescaled GHP-16 scores were summed across all 16 items, yielding a total score ranging from 16 to 80, in accordance with international reporting practices.

Example items from the GHP-16 include the following:“I go regularly to the doctor for a check-up”.“I collect information about things that concern my health”.“I let myself be vaccinated”.

These reflect the typical domains covered by the scale, such as preventive care utilization, health information seeking, and vaccination behavior.

Consistent with the scoring method used in the international literature, the GHP-16 total score was computed as the arithmetic mean across all 16 items, yielding a continuous scale from 1.0 (very low engagement in health-promoting behavior) to 5.0 (very high engagement). This average score approach is widely used and allows for direct comparisons across populations. The item scores were not reverse-coded because all items were phrased positively. For participants with one or two missing responses, the mean was calculated based on the available items, and cases with more than two missing items were excluded from the total score analyses. The internal consistency of the scale in our sample was high (Cronbach’s α = 0.77).

### 2.4. Health Literacy and Patient Activation

Health literacy was assessed using the 16-item European Health Literacy Survey Questionnaire (HLS-EU-Q16), which measures the perceived ease or difficulty in accessing, understanding, appraising, and applying health-related information [13,14,15]. Each item was rated on a four-point Likert scale and scored according to validated procedures, yielding a summary index ranging from 0 to 16, with higher scores indicating better health literacy. In line with established scoring protocols, participants were required to respond to at least 14 of the 16 items to calculate the index. Respondents who answered fewer than 14 items (21.1% of the weighted sample) were categorized as “missing” and treated as a separate group in all descriptive and inferential analyses. This approach allowed the inclusion of these cases in the regression models while preserving transparency in the handling of incomplete data.

The representative items included:“How easy is it for you to understand what your doctor says to you?”.“How easy is it to judge whether the health information in the media is reliable?”.

These items assess the perceived comprehension, appraisal, and application of health information.

Patient activation, reflecting on an individual’s knowledge, skills, and confidence in managing their health, was assessed using the PAM-10 [16,17,18]. Scores were transformed to a standardized 0–100 scale in accordance with the scoring guidelines, where higher values denoted greater activation. PAM-10 scores were categorized using standard thresholds: Level 1 (“disengaged and overwhelmed”), Level 2 (“becoming aware but struggling”), Level 3 (“taking action”), and Level 4 (“maintaining behavior”).

Example items include the following:“I am confident that I can take actions that will help prevent or minimize some symptoms or problems associated with my health condition”.“I am confident that I can tell a doctor concerns I have even when he or she does not ask”.

These items capture the respondents’ confidence, self-management capacity, and communication behavior.

### 2.5. Mistrust in Health Information

To assess skepticism toward formal medical advice, a Mistrust Index was constructed using four items querying perceived trust in health information provided by general practitioners, outpatient specialists, pharmacists, and nurses, as previously described [10]. Each item was rated on a 4-point Likert scale, ranging from 1 (“very trustworthy”) to 4 (“not at all trustworthy”). Item responses were summed to yield an index score between 4 and 16, with higher scores denoting greater mistrust of professional sources of health information. The internal reliability of the index resulted in a high Cronbach’s α of 0.76.

### 2.6. Statistical Analyses

The analysis workflow combined several open-source Python packages, including pandas (v2.2.2) for data handling [19], NumPy (v1.26.4) for numerical computations [20], SciPy (v1.13.0) for statistical testing and correlation analysis [21], statsmodels (v0.14.0) for regression modeling and confidence interval estimation [22], and matplotlib (v3.8.0) for data visualization [23]. All codes and documentation are available in a dedicated GitHub repository to support reproducibility (see Data Availability Statement). All statistical analyses were performed and validated using Python version 3.10.12 scripts executed in Google Colab with version-controlled package.

Descriptive statistics were calculated for all relevant variables, including the means and standard deviations for continuous variables and proportions for categorical variables. Group comparisons of the GHP-16 total scores across categorical variables were performed using Mann–Whitney U tests for binary variables and Kruskal–Wallis tests for variables with more than two groups. Weighted medians and 95% confidence intervals were calculated using bootstrapping. Effect sizes were reported as rank-biserial correlation coefficients (*r*_rb_) for binary comparisons and epsilon squared (ε^2^) for multi-group comparisons. Given the exploratory nature of the subgroup comparisons, no correction for multiple tests was applied. Interpretation focused on effect size estimates and confidence intervals rather than *p*-values.

Associations between health behavior scores and key predictors (e.g., health literacy, patient activation, and Mistrust Index) were assessed using Spearman’s rank correlation and multivariable linear regression. The GHP-16 score was used as a continuous dependent variable.

#### Regression Modeling and Covariate Selection

To assess the independent associations between mistrust in professional health information, health literacy, patient activation, and engagement in preventive health behaviors, weighted linear regression (WLR) models were estimated using the GHP-16 total score as a continuous outcome variable. Two models were specified: one examining all predictors simultaneously and one focusing specifically on the contribution of the language group after adjusting for potential confounders.

All models included sociodemographic covariates (age, gender, education, living situation, language group, and employment in the health or social care sector) and categorical measures of health literacy (HLS-EU-Q16) and patient activation (PAM-10). Health literacy was modeled using three substantive categories (inadequate, problematic, and sufficient) and a fourth category for missing responses. Patient activation was entered as a four-level ordinal variable (levels 1–4), using the lowest level (“disengaged and overwhelmed”) as the reference. Covariates were selected a priori based on their theoretical relevance and prior empirical evidence. All theoretically relevant variables were included simultaneously in the multivariable models to ensure full adjustment for potential confounders. No variable selection procedures (e.g., stepwise regression) were used.

The overall explanatory power of the regression model was assessed using the adjusted coefficient of determination (adjusted R^2^), which indicates the proportion of variance in preventive health behavior explained by the included predictors. Multicollinearity was assessed using variance inflation factors (VIFs), with all predictors exhibiting VIF values well below the conventional thresholds (VIF < 2), indicating no collinearity concerns. Given the large sample size (*n* = 2090) and favorable observation-to-variable ratio (>200:1), the risk of overfitting was negligible. The regression results are reported as unstandardized beta coefficients (ß) with 95% confidence intervals.

## 3. Results

### 3.1. Variation in Health-Promoting Behavior Across Sociodemographic and Psychosocial Characteristics

As shown in Table 1, the total GHP-16 scores varied significantly across most sociodemographic, health, and psychosocial characteristics in the adult population of South Tyrol. Statistically significant differences (*p* < 0.001) were observed for gender, age group, language, residence, education, health literacy, patient activation, and self-rated health. Additional significant variation was found for working in the health or social sector (*p* < 0.001) and for the presence and number of chronic conditions (*p* = 0.001 and *p* = 0.022, respectively).

Effect size estimates indicated that gender was the strongest predictor of preventive health behavior, with women reporting substantially higher GHP-16 scores than men (rank-biserial correlation *r*_rb_ = 0.597). Moderate effects were observed for employment in the health or social care sector, with higher preventive behavior among employees in this sector. Preventive behavior also increased with age, with the highest GHP-16 scores among individuals aged 55 and older and the lowest among respondents aged 18–34. Likewise, patient activation showed a positive gradient, with higher scores in the upper PAM-10 tiers, reflecting greater engagement in health-promoting practices.

Most of the remaining variables, including language, education, and self-rated health, were associated with small-to-moderate effect sizes, supporting the presence of consistent but less pronounced behavioral differences across subgroups. Notably, respondents with missing health literacy data reported the lowest GHP-16 scores, suggesting that non-response to the HLS-EU-Q16 may be a marker of low engagement in preventive health behavior.

These findings underscore the relevance of demographic, educational, and motivational factors, particularly gender, age, and patient activation, in shaping health behavior patterns in the general population.

### 3.2. Associations of Mistrust, Health Literacy, Patient Activation, and Sociodemographic Predictors with Preventive Health Behavior

Figure 2 illustrates the relationship between the Mistrust Index and GHP-16 total score. A downward trend is evident, with higher levels of mistrust consistently associated with lower engagement in health-promoting behavior. This association was statistically significant (*p* < 0.001). According to Cohen’s guidelines, the effect size corresponds to a small but practically relevant inverse relationship between mistrust and health behavior. This consistent trend supports the interpretation that skepticism toward medical professionals is negatively related to preventive health practices in the general population. Given the population weights applied, these findings are generalizable to the adult population of South Tyrol.

Table 2 presents the results of the regression analyses. In the weighted linear regression model, higher levels of mistrust in health information from professional sources were independently associated with lower engagement in preventive health behaviors, even after adjusting for sociodemographic characteristics, health literacy, and patient activation. The model explained 25.2% of the variance in preventive health behavior scores (R^2^ = 0.252), and the adjusted R^2^ was 0.246, accounting for the number of covariates. No multicollinearity was detected among the independent variables (all VIFs < 2), which supported the stability and interpretability of the regression estimates. The negative association between mistrust and the GHP-16 score was small in absolute magnitude but statistically significant and consistent in its direction.

The sociodemographic covariates retained their expected associations. Female respondents and older individuals reported higher engagement in preventive health behaviors. A clear educational gradient was observed: respondents with higher formal education reported substantially higher GHP-16 scores than those with lower educational attainment. Italian speakers also reported higher scores than German speakers, whereas respondents from other language groups did not differ significantly.

Neither living alone nor working in the health or social care sector was independently associated with preventive behavior after adjustments.

Health literacy was positively and significantly associated with the GHP-16 score. Although the difference between problematic and inadequate literacy was not statistically significant, respondents with sufficient health literacy reported notably higher preventive behavior. Interestingly, respondents with missing HLS-EU-Q16 responses did not differ significantly from the reference group, although their lower point estimates aligned with the descriptive findings.

Patient activation exhibited the strongest graded association with health behaviors in the model. With each higher level of activation, the GHP-16 scores increased substantially, with particularly large differences observed for those in the highest activation tier.

Together, these findings support the relevance of both attitudinal (mistrust) and capability-based (literacy and activation) factors in shaping engagement in health-promoting practice.

### 3.3. Health-Promoting Behavior by Language Group: German vs. Italian

As illustrated in Figure 3, comparisons of individual GHP-16 items revealed systematic, albeit modest, differences in preventive health behavior between the language groups. Italian-speaking respondents reported higher mean scores than German-speaking respondents on nearly all items. The visual layout highlights this pattern, with Italian means consistently extending further to the right and German means plotted symmetrically to the left.

While many of the differences achieved statistical significance (*p* < 0.05), most were associated with negligible effect sizes, according to Cliff’s delta (*d* < 0.147). Small effects were observed for three behaviors: taking dietary supplements (*d* = 0.22), gathering health information (*d* = 0.20), and receiving vaccinations (*d* = 0.16), all of which favored Italian speakers. These items may reflect differences in health-seeking behavior or informational engagement across linguistic and cultural groups. The largest effect was observed for supplement use, which was both statistically significant (*p* < 0.001) and within the small-effect range.

Other behaviors, such as flossing, regular dental or medical checkups, brushing teeth, and avoiding smoking, showed virtually identical engagement levels between the groups (*d* ≈ 0). Notably, no item reached the medium or large effect size threshold, underscoring the limited practical magnitude of these differences despite their statistical detectability.

To evaluate whether language group differences in preventive health behaviors persist independently of mistrust attitudes, a second model was estimated, excluding the Mistrust Index. This model centered on language group, health literacy, and patient activation as explanatory variables. The goal was to clarify whether linguistic disparities remain relevant when institutional mistrust is omitted from the analysis. Table 3 and Appendix A present the results of the WLR model estimating GHP-16 total scores as a function of language group, sociodemographic characteristics, health literacy, and patient activation. All variables were entered simultaneously to estimate their independent associations with health-promoting behaviors. The model explained 24.7% of the variance in preventive behavior scores (R^2^ = 0.247), and the adjusted R^2^ was 0.242 after accounting for the number of predictors.

Italian speakers had significantly higher GHP-16 scores than German speakers (β = 0.377, 95% CI [0.172; 0.581], *p* < 0.001), even after adjusting for covariates. This suggests that the behavioral advantage observed at the descriptive level persists even after accounting for demographic and psychosocial differences. The estimate for speakers of other languages was negative but not statistically significant.

Female sex, older age, and higher education level were significantly associated with increased engagement in preventive behaviors. Compared to individuals with middle school or lower education, those with vocational training, high school, and university degrees showed progressively higher scores, with the strongest effect observed among university graduates (β = 1.188, 95% CI [0.904, 1.473], *p* < 0.001). Health literacy was also associated with health behaviors in a graded manner. Compared with respondents with problematic literacy, those with sufficient literacy had higher GHP-16 scores (β = 0.548, 95% CI [0.268; 0.828], *p* < 0.001), while the estimate for inadequate literacy was modest and borderline significant. Respondents with missing HLS-EU-Q16 data did not differ significantly from the reference groups. Patient activation emerged as a strong graded predictor of preventive behavior. Compared to the least activated group, individuals in each higher activation tier reported increasingly higher GHP-16 scores, with the highest effect observed in the “maintaining” group (β = 1.739, 95% CI [1.439, 2.039], *p* < 0.001).

Other covariates, including living alone and employment in the health or social sector, were not significantly associated with the outcomes after adjustment.

## 4. Discussion

This population-based study examined the distribution and determinants of health-promoting behavior in South Tyrol using a validated GHP-16 scale. Preventive behaviors varied significantly across sociodemographic and psychosocial strata, with higher engagement observed among women, older adults, and those with higher education levels. Patient activation emerged as the strongest independent predictor of health behavior, followed by that of health literacy. The mistrust of professional health information was inversely associated with preventive practices, although the effect size was modest. Italian-speaking respondents reported higher engagement in several preventive behaviors, a difference that persisted after full adjustment for sociodemographic and psychosocial factors of the respondents.

These findings underscore the combined influence of demographic, attitudinal, and capability-based factors in shaping health behavior and highlight the importance of the linguistic-cultural context in multilingual regions. The observed associations align with the HBM. Individuals with higher health literacy and activation may perceive greater benefits and confidence, reflecting the HBM domains of perceived benefits and self-efficacy. Higher mistrust acts as a perceived barrier, whereas language group differences may reflect cultural cues that shape behavioral readiness. This supports the applicability of the HBM in multilingual and diverse populations.

### 4.1. Interpretation in Light of the Existing Literature

Several studies have consistently demonstrated that higher patient activation, as measured by the Patient Activation Measure (PAM-10 or PAM-13), is strongly associated with greater engagement in preventive health behaviors in the general adult population. Individuals with higher PAM scores are more likely to engage in activities such as healthy eating, regular exercise, screening adherence, and proactive self-care behaviors [24,25,26]. Moreover, changes in activation levels can predict corresponding changes in preventive behavior and clinical outcomes, including lower healthcare costs and improved biomedical indicators [26,27]. Psychosocial factors, such as self-efficacy, emotional support, and high-quality patient–provider communication, are known to enhance patient activation, which, in turn, mediates engagement in preventive practices [17,25,28]. These findings align closely with our results, where patient activation emerged as the strongest independent predictor of health-promoting behaviors.

Consistent with prior research, our findings confirm that gender, age, and education are important determinants of preventive health behavior. Women tend to engage more frequently in preventive practices, such as health screenings and checkups, a pattern linked to differences in health awareness and information-seeking behavior [29,30]. Older adults also show higher engagement, particularly in behaviors such as routine medical visits and screening adherence, likely reflecting increased health needs and more frequent interactions with healthcare providers [30,31]. Higher educational attainment is associated with a greater uptake of preventive measures, supporting the evidence that education enhances the ability to access, interpret, and apply health information [30,32,33]. These gradients were reproduced in our adjusted models and underscore the relevance of structural and behavioral determinants in shaping population-level preventive engagement.

The mistrust of professional health information has been identified as a significant barrier to engaging in preventive health behaviors, particularly among marginalized populations. Our findings confirm a small but consistent inverse relationship between mistrust and self-reported preventive behaviors. This aligns with prior evidence linking higher medical mistrust to delays in routine screening and lower vaccination uptake [34,35]. Discrimination experiences, both personal and vicarious, further amplify mistrust and contribute to behavioral disengagement [36]. Additionally, low health literacy is an independent predictor of institutional mistrust, compounding the risk of nonengagement among vulnerable groups [37]. Although some studies have reported mixed findings, the overall pattern suggests that trust in the medical system is a critical determinant of preventive action. Public health strategies aimed at improving preventive behavior may benefit from explicitly addressing sources of mistrust, alongside informational and structural barriers.

The findings confirm that higher health literacy is significantly associated with greater engagement in preventive health behaviors, independent of sociodemographic and activation-related factors. This is consistent with international evidence showing that adults with higher health literacy are more likely to participate in screenings, adopt healthy lifestyles, and adhere to medical recommendations [38,39,40,41]. Among older adults, adequate health literacy has been linked to increased participation in mammography, physical activity, and reduced tobacco use [38,41]. Similar associations have been documented in patients with chronic diseases, where health literacy predicts healthier diets, better disease management, and improved self-reported health [39]. Population-based studies in Korea and China further demonstrate that health literacy, along with knowledge and media literacy, is a significant determinant of preventive behavior, particularly in pandemic contexts [28,40]. These consistent patterns underscore the relevance of both individual- and system-level health literacy as drivers of preventive engagement across diverse populations.

Regional cultural and linguistic contexts, such as those found in bilingual settings like South Tyrol, play a meaningful role in shaping preventive health engagement and trust in the healthcare system. Linguistic minorities often experience health communication anxiety when using a second language with providers, which can reduce service uptake, especially for sensitive or preventive care [42]. Structural limitations, including the lack of services in minority languages, further exacerbate these barriers [43]. Conversely, the availability of bilingual staff and culturally adapted materials has been shown to improve engagement in preventive programs, particularly among immigrant and minority populations [44,45]. Cultural sensitivity at both the surface (language) and deep (content validation) levels enhances participation in health interventions [46]. Beyond individual and service-level factors, regional policies and community attitudes toward bilingualism also influence whether linguistic identity is experienced as a barrier or resource, thereby shaping institutional trust and health behavior [47].

### 4.2. Language Group Differences and Cultural Implications

These broader findings help to contextualize the persistent differences observed in our study between Italian- and German-speaking participants. In South Tyrol, a region with entrenched bilingual structures and distinct linguistic communities, such patterns may reflect both cultural orientation and systemic access.

The behavioral differences observed in this study, specifically higher levels of health-promoting behavior among Italian-speaking compared to German-speaking adults in South Tyrol, persisted even after full adjustment for sociodemographic and psychosocial variables. These differences are consistent with earlier findings from South Tyrol, showing higher vaccine hesitancy, lower institutional trust, and greater reliance on alternative medicine among German speakers [7,8,9].

The international literature supports the view that such differences may reflect more than just individual characteristics. In multilingual regions, language and cultural mismatches are known to hinder trust and engagement with health services [48,49,50]. Minority language speakers often report discomfort or anxiety during clinical interactions, especially when care is not provided in their preferred language [49,51]. Structural barriers, such as the lack of culturally adapted materials, language-concordant providers, or community-tailored outreach, can further reduce engagement in prevention [52,53].

South Tyrolean studies confirm that information sources, communication preferences, and trust levels differ markedly among linguistic groups, even within the same health system [10,54]. These findings suggest that, beyond formal bilingualism, deeper cultural and communicational factors may shape how health messages are received and acted upon. Interventions relying solely on translated materials or system-wide policies may fall short if they do not engage with group-specific trust dynamics and social identity.

In addition to individual- and system-level factors, variations in exposure to national versus regionally coordinated health promotion campaigns may also play a role. Given that Italian-speaking residents may engage more frequently with Italian national media, whereas German-speaking residents may rely more on local or German-language sources, differences in messaging exposure, framing, and campaign intensity could influence preventive behavior patterns between the two groups.

Considering these insights, the behavioral gap observed between Italian and German speakers in this study likely reflects not only differential access or literacy but also culturally embedded orientations toward health systems and professional information. Addressing such disparities requires not only linguistic equity in service delivery but also the co-development of culturally resonant health communication strategies grounded in trust and tailored to distinct identity groups [52,55,56].

### 4.3. Strengths and Limitations

This study has several strengths. First, it is based on a stratified, population-representative sample of adults in South Tyrol, allowing weighted estimates that reflect the region’s demographic structure. Second, validated and internationally recognized instruments were employed to assess preventive health behaviors (GHP-16), health literacy (HLS-EU-Q16), and patient activation (PAM-10), ensuring conceptual clarity and measurement reliability. Third, the analyses were adjusted for a comprehensive set of sociodemographic and psychosocial covariates, and model diagnostics confirmed the robustness of the results.

However, this study has several limitations that must be acknowledged. The cross-sectional design precludes causal inference and limits the interpretation of directional relationships between predictors and health behavior. All data were self-reported, introducing the potential for measurement error, including social desirability and recall bias. The GHP-16 items were translated into German and Italian by the ASTAT team, using standard translation procedures. However, formal psychometric validation of the translated versions has not been conducted. Therefore, the observed differences between the language groups may, in part, reflect language- or culture-specific interpretation effects. Although care was taken to ensure clarity and neutrality in item wording, the use of a four-point Likert scale, which was later rescaled to match the original five-point GHP-16 format, may have reduced response variability or comparability with prior studies.

Treating missing responses on the HLS-EU-Q16 as a separate category may have conflated non-response with low health literacy. While this approach preserved the sample size, it may have underestimated the strength of the associations with actual literacy levels. As preventive health behaviors were self-reported, the potential for social desirability bias must be acknowledged in this study. Respondents may have overreported their engagement in health-promoting activities, which could partly account for the generally high GHP-16 scores observed. Furthermore, while the sample is representative of South Tyrol, generalizability to other multilingual or nationally centralized health systems may be limited because of the region’s unique sociolinguistic and institutional context. Finally, despite multivariable adjustment, residual confounding could not be fully excluded.

### 4.4. Public Health and Policy Implications

The findings of this study highlight important ways to improve public health and regional prevention planning. Patient activation and health literacy are the most influential psychosocial predictors of engagement in preventive health behaviors. Strengthening these capacities through tailored communication, shared decision making, and skill-building interventions should be central to efforts aimed at promoting behavior change at the population level [57].

In multilingual regions such as South Tyrol, persistent behavioral differences between language groups underscore the necessity of language-sensitive strategies. Public health messaging and prevention programs should be developed and implemented with careful attention to linguistic preferences, cultural norms, and group-specific trust dynamics. This includes not only the use of bilingual materials and services but also the co-design of interventions with members of both linguistic communities to ensure relevance and accessibility [58].

The population-representative nature of this study and its focus on modifiable psychosocial determinants make its findings directly applicable to regional prevention frameworks. The results may inform the targeting of health promotion efforts under the South Tyrolean prevention plan, with particular emphasis on reaching groups with lower activation, insufficient health literacy, and linguistic-specific engagement barriers.

The observed associations in this study align with the core constructs of the HBM (Figure 1). Patient activation and health literacy reflect perceived self-efficacy and control over health, whereas mistrust in professional health information functions as a perceived barrier to action. These findings reinforce the relevance of HBM-guided strategies for behavioral intervention design, including enhancing confidence in one’s ability to manage health, increasing the perceived benefits of preventive actions, and reducing informational and relational barriers to engagement. Thus, the HBM provides a valuable framework for shaping culturally and linguistically responsive prevention strategies in multilingual regions.

The results also highlight opportunities for nursing professionals to play a central role in strengthening patient activation and health literacy, particularly through linguistically and culturally adapted communication strategies. Community and public health nurses are well-positioned to deliver trust-building interventions that address mistrust and promote engagement in preventive practices, especially among minority language groups.

The findings must be interpreted by considering South Tyrol’s distinctive sociolinguistic and political structures. As a decentralized, bilingual province with strong local governance and region-specific health policies, the dynamics observed here, particularly those related to language, trust, and preventive behavior, may not be extrapolated to nationally centralized or culturally homogeneous contexts. Comparative research in other multilingual or minority language settings is needed to assess the broader relevance of these findings.

## 5. Conclusions

This population-based study provides new evidence on the distribution and determinants of preventive health behaviors in South Tyrol, a linguistically diverse region in Northern Italy. Using the validated GHP-16 scale, preventive behavior varied significantly across sociodemographic and psychosocial strata. Gender, age, and education emerged as strong predictors, along with two modifiable psychosocial constructs: patient activation and health literacy. Among these, patient activation demonstrated the strongest independent association with preventive health behaviors, showing a clear dose–response relationship across activation levels. Health literacy was also positively related to behavior, particularly when functional skills were rated as sufficient.

Persistent differences in preventive behaviors were observed between Italian- and German-speaking respondents, even after adjusting for relevant covariates. These differences likely reflect culturally embedded orientations toward the health system and varying degrees of trust in professional health information sources. Language group affiliation may thus serve as a marker for deeper communication and cultural patterns that influence health behavior.

These findings support the prioritization of patient activation and health literacy as key intervention targets for the promotion of preventive health behaviors. In multilingual regions, such as South Tyrol, communication strategies should further account for linguistic preferences, cultural norms, and trust dynamics. Bilingual service provision, culturally adapted health promotion, and participatory intervention design are critical for ensuring equitable access and engagement.

These results offer relevant insights for prevention planning in decentralized and culturally pluralistic health systems, emphasizing the need for socially and linguistically inclusive strategies in public health promotion.

## Figures and Tables

**Figure 1 nursrep-15-00240-f001:**
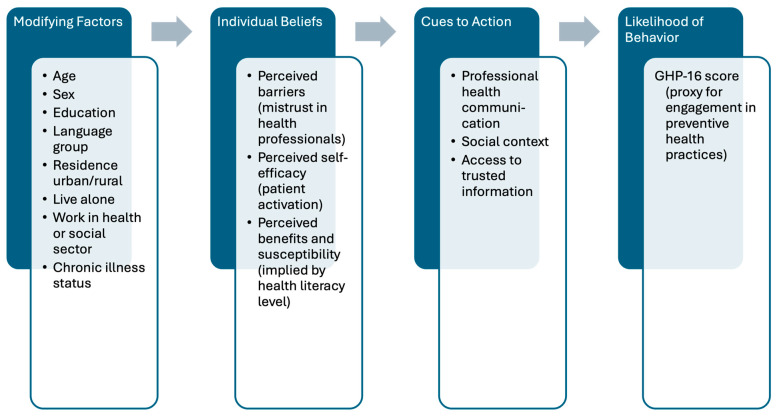
Conceptual model linking health literacy, mistrust, and health behavior based on the health belief model.

**Figure 2 nursrep-15-00240-f002:**
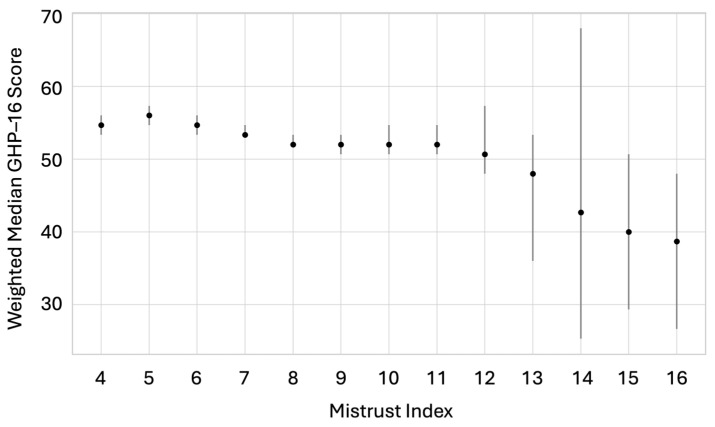
Weighted median GHP-16 score by the level of mistrust in professional health information (*n* = 2090; weighted). Vertical lines indicate the 95% confidence intervals (CIs) obtained by bootstrapping. Higher index values correspond to greater mistrust in professional health information sources (general practitioners, specialists, pharmacists, and nurses). Spearman’s ρ = −0.20 (*p* < 0.001).

**Figure 3 nursrep-15-00240-f003:**
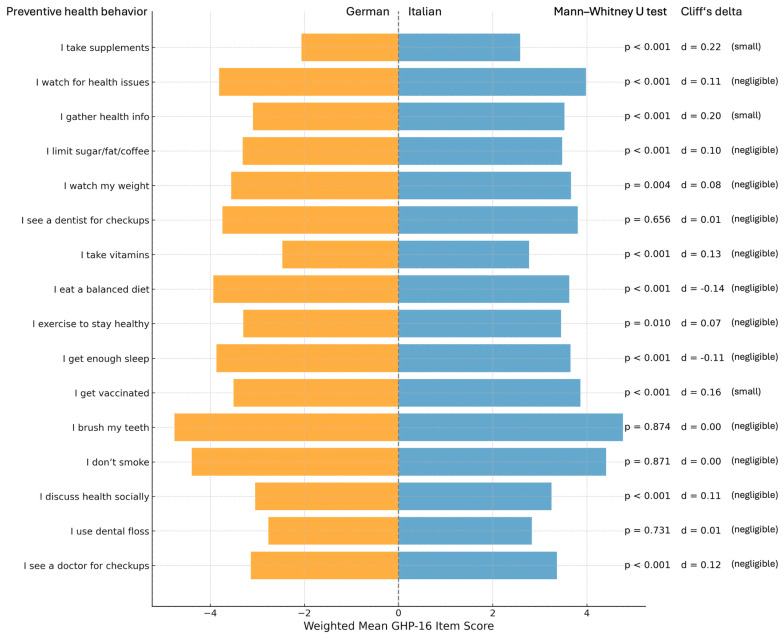
Weighted mean scores for each GHP-16 item, stratified by language group (German Vs. Italian speakers) in South Tyrol. Bars represent mean scores; German-speaking participants are plotted in the negative direction for visual symmetry. Scores range from 1 (never) to 5 (always) and reflect the frequency of engaging in each preventive behavior. For each item, the *p*-value from the Mann–Whitney U test and Cliff’s delta (d) are displayed to indicate statistical and practical significance, respectively. Interpretation of Cliff’s delta: negligible (*d* < 0.147), small (*d* < 0.33), medium (*d* < 0.474), and large (*d* ≥ 0.474).

**Table 1 nursrep-15-00240-t001:** Distribution of the GHP-16 total score by sociodemographic, health, and psychosocial characteristics in the adult population of South Tyrol (N = 2090; weighted).

Variable	Category	*n* (Weighted)	GHP-16 Score ^1^			Comparison	
			Median	95% CI Lower ^2^	95% CI Upper ^2^	*p*-Value ^3^	Effect Size ^4^
Gender	Male	1026	50.7	50.67	50.70	<0.001	0.597
	Female	1064	53.3	54.67	56.00		
Age group	18–34 years	496	50.7	49.33	52.00	<0.001	0.026
	35–54 years	683	50.7	52.00	53.33		
	55–99 years	912	56.0	54.67	56.00		
Residence	Rural	1665	50.7	53.33	53.33	0.001	−0.117
	Urban	425	53.3	53.33	56.00		
Education	Middle school	441	50.7	52.00	53.33	<0.001	0.032
	Vocational school	666	49.3	50.67	52.00		
	High school	545	52.0	53.33	54.67		
	University	1885	50.7	53.33	53.33		
Citizenship	Italian	205	53.3	49.33	56.00	0.559	0.071
	Other	205	53.3	34.47	68.07		
Language	German	1330	50.7	52.00	53.33	<0.001	0.015
	Italian	480	53.3	54.67	56.00		
	Other	280	50.7	49.33	54.67		
Lives alone	No	1718	52.0	53.33	53.33	0.545	0.005
	Yes	372	52.0	53.33	54.67		
Works in health or social sector	No	1874	50.7	53.33	53.33	<0.001	−0.353
	Yes	215	54.7	54.67	57.33		
Self-rated health status	High	1447	54.7	37.33	69.33	0.001	0.010
	Middle	511	53.3	36.00	69.33		
	Low	131	53.3	28.57	67.43		
Health literacy (HLS-EU_Q16)	Inadequate	266	50.7	50.67	53.33	<0.001	0.052
	Problematic	559	52.0	52.00	53.33		
	Sufficient	824	53.3	54.67	56.00		
	Unknown/missing	441	48.0	49.33	52.00		
Patient activation (PAM-10)	Disengaged and overwhelmed	336	49.3	49.33	50.67	<0.001	0.079
	Becoming aware but still struggling	888	50.7	52.00	53.33		
	Taking action	490	50.7	53.33	54.67		
	Maintaining behaviors and pushing further	377	56.0	57.33	60.00		
Chronic disease	No	1357	50.7	52.00	53.33	0.001	−0.171
	Yes	733	53.3	53.33	54.67		
Number of chronic diseases	1	496	53.3	53.33	54.67	0.022	0.006
	2	162	53.3	53.33	54.67		
	3	60	52.0	52.00	56.00		
	4	10	53.3	48.00	61.33		
	5	6	53.3	44.00	58.67		

^1^ GHP-16 score: Sum of 16 rescaled items (range: 16–80), with higher scores indicating greater engagement in health-promoting behavior. ^2^ 95% CI: Bootstrap-based confidence intervals for weighted median. ^3^ *p*-values: Mann–Whitney U tests (two groups) or Kruskal–Wallis tests (≥three groups). ^4^ Effect sizes: Rank-biserial correlation (*r*_rb_) for two-group comparisons; epsilon squared (ε^2^) for multi-group comparisons. Positive values indicated higher scores in the reference group. Interpretation thresholds: *r*_rb_ ≈ 0.10 = small, ≈0.30 = moderate, ≥0.50 = large; ε^2^ ≈ 0.01 = small, ≈0.06 = moderate, and ≥0.14 = large.

**Table 2 nursrep-15-00240-t002:** Multivariable linear regression predicting GHP-16 preventive health behavior scores based on the Mistrust Index, HLS-EU-Q16, and PAM-10 scores and sociodemographic variables (*n* = 2090; weighted).

Variable	Regression Coefficient ß	95% CI [Lower; Upper]	Std. Error	t-Statistic	*p*-Value
const	6.532	[5.885; 7.178]	0.33	19.818	<0.001
Mistrust_Index	−0.05	[−0.09; −0.009]	0.021	−2.42	0.016
Female (vs. Male)	1.183	[1.018; 1.347]	0.084	14.097	<0.001
Age (in years)	0.025	[0.02; 0.030]	0.003	9.306	<0.001
Education (vs. middle school or lower)					
Vocational school	0.363	[0.135; 0.592]	0.116	3.119	0.002
High school	0.737	[0.488; 0.987]	0.127	5.798	<0.001
University	1.130	[0.855; 1.406]	0.14	8.046	<0.001
Language (vs. German)					
Italian	0.372	[0.177; 0.567]	0.1	3.738	<0.001
Other	−0.193	[−0.48; 0.095]	0.147	−1.312	0.190
Lives alone (vs. no)	−0.130	[−0.339; 0.088]	0.107	−1.214	0.225
Works in health or social sector (vs. no)	0.016	[−0.258; 0.291]	0.14	0.117	0.907
HLS-EU-Q16 (vs. problematic) ^1^					
Inadequate	0.237	[−0.046; 0.520]	0.144	1.645	0.100
Sufficient	0.518	[0.238; 0.798]	0.143	3.627	<0.001
Missing/unknown	−0.126	[−0.423; 0.172]	0.152	−0.827	0.408
PAM-10 (vs. disengaged and overwhelmed) ^1^					
Becoming aware	0.603	[0.360; 0.846]	0.124	4.86	<0.001
Taking action	0.875	[0.603; 1.147]	0.139	6.315	<0.001
Maintaining	1.61	[1.319; 1.902]	0.149	10.823	<0.001

^1^ Categorical definitions of health literacy and patient activation levels are described in the Methods Section. CI, Confidence interval; HLS-EU-Q16, Health Literacy Score EU Questionnaire 16 items; and PAM-10, Patient Activation Measure 10 items.

**Table 3 nursrep-15-00240-t003:** Multivariable linear regression predicting GHP-16 preventive behavior scores, focusing on linguistic group differences, adjusted for sociodemographic and psychosocial covariates (*n* = 2090; weighted).

Variable	Regression Coefficient ß	95% CI [Lower; Upper]	Std. Error	t-Statistic	*p*-Value
const	5.614	[5.065; 6.164]	0.28	20.038	<0.001
Female (vs. male)	1.239	[1.069; 1.410]	0.087	14.283	<0.001
Age (in years)	0.026	[0.021; 0.032]	0.003	9.725	<0.001
Lives alone (vs. no)	−0.156	[−0.374; 0.061]	0.111	−1.411	0.158
Language (vs. German)					
Italian	0.377	[0.172; 0.581]	0.104	3.614	<0.001
Other	−0.249	[−0.542; 0.044]	0.149	−1.664	0.096
Education (vs. middle school or lower)					
Vocational school	0.398	[0.162; 0.634]	0.12	3.309	0.001
High school	0.828	[0.569; 1.087]	0.132	6.272	<0.001
University	1.188	[0.904; 1.473]	0.145	8.198	<0.001
Works in health or social sector (vs. no)	0.028	[−0.255; 0.311]	0.144	0.194	0.846
HLS-EU-Q16 (vs. problematic) ^1^					
Inadequate	0.321	[0.037; 0.606]	0.145	2.215	0.027
Sufficient	0.548	[0.268; 0.828]	0.143	3.84	<0.001
Missing/unknown	−0.19	[−0.488; 0.108]	0.152	−1.252	0.211
PAM-10 (vs. disengaged and overwhelmed) ^1^					
Becoming aware	0.657	[0.413; 0.900]	0.124	5.29	<0.001
Taking action	1.011	[0.737; 1.284]	0.14	7.241	<0.001
Maintaining	1.739	[1.439; 2.039]	0.153	11.353	<0.001

Results from a weighted linear regression model using the GHP-16 total score as a continuous outcome variable. Language group (Italian vs. German [ref. German]) along with age, gender, education, living situation, employment in the health or social sector, health literacy (HLS-EU-Q16), and patient activation (PAM-10). ^1^ Categorical definitions of health literacy and patient activation levels are described in the Methods Section. The coefficients (ß) represent the mean difference in the GHP-16 score associated with each category or unit increase while holding all other variables constant. The 95% confidence intervals (CI) and *p*-values are reported.

## Data Availability

The original data presented in this study are openly available in GitHub at https://github.com/wiedermc/ghp16-health-behavior.git (accessed on 20 June 2025).

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
