# Peer review of "Preventive Health Behavior and Readiness for Self-Management in a Multilingual Adult Population: A Representative Study from Northern Italy"

_nursrep, 2025, doi:10.3390/nursrep15070240_

Round 1
Reviewer 1 Report
Comments and Suggestions for Authors
The manuscript titled "Preventive Health Behavior and Readiness for Self-Management in a Multilingual Adult Population: A Representative Study from Northern Italy" offers valuable and timely insights into health-related behaviors within a culturally diverse European context. The study is grounded in a strong methodological framework and features a thorough analytical approach. However, a few aspects may need further clarification to improve understanding and quality.
The Introduction would benefit from some tightening, as certain ideas—like the study aims—are repeated in slightly different ways in a few places (for example, around lines 70–81). Clarifying and unifying these parts could make the narrative easier to follow.
Although the study is well-conceived, the manuscript should more clearly highlight its contribution beyond existing research, especially in relation to previous findings from South Tyrol.
Although the HBM is referenced as a theoretical framework, its practical application in the study is somewhat limited. Clarifying how the study variables reflect specific HBM constructs—such as perceived barriers or cues to action—would strengthen the theoretical foundation.
It’s unclear whether the German and Italian versions of the GHP-16 were psychometrically validated, which raises questions about the reliability of cross-language comparisons. If validation was done, it should be described more clearly; if not, this should be acknowledged as a limitation.
Have the authors considered whether treating missing responses on the HLS-EU-Q16 as a separate category might conflate absence of data with absence of health literacy? Might a strategy like multiple imputation offer a more accurate handling of missing data?
As health behaviors are self-reported, the risk of social desirability bias is high. This limitation is not discussed but should be addressed, especially given the high engagement scores reported.
While the study is representative for South Tyrol, the unique sociolinguistic and institutional context constrains the applicability of results to other multilingual or centralized health systems. This point should be more explicitly discussed.
The phrase “language group affiliation reflects deeper communicational influences” feels quite vague and could be made clearer. It would help to use more specific, evidence-based language—for example, suggesting that language group might reflect differences in trust in information sources or access to health information.
Despite referencing validated scales, the manuscript does not provide any example items. Including 1–2 representative items per instrument (e.g., "I exercise regularly") would improve transparency and reader understanding of what behaviors or attitudes were measured.
Reviewer 2 Report
Comments and Suggestions for Authors
The authors have invested considerable effort in conducting a well-designed and presented study on preventive health behavior in a culturally and linguistically diverse region.
Minor suggestions:
1. In the Limitations section, I recommend further emphasizing that the GHP-16 scale was used in translated versions (Italian and German), and that formal psychometric validation of these versions was not performed — this may partly contribute to the observed language group differences.
2. The potential influence of self-report bias (social desirability) on preventive behavior outcomes could also be more explicitly discussed.
Other comments are in the attachment
Overall, this is a high-quality manuscript.

Reviewer 3 Report
Comments and Suggestions for Authors
The manuscript provides the findings of a study conducted with the objective to examine how sociodemographic, psychosocial, and linguistic factors shape preventive behavior patterns of a sample of the adult population of South Tyrol, an autonomous and bilingual province in northern Italy. The manuscript addresses current and relevant issues associated with the self-management of chronic disease. There is a high component of originality in the study. The theoretical basis is well founded and justifies the study. The objective is relevant and clearly defined. The methods used are sufficiently described, appropriate and consistent with the study proposal. The results are presented in an attractive manner, and the discussion of the findings show excellent prospects for the area of knowledge, which enhances the quality of the manuscript. Bibliographic references are current and relevant to the subject of study.
Author Response
We sincerely thank Reviewer 3 for the thoughtful and encouraging evaluation of our manuscript. We are especially grateful for the recognition of the study’s originality, relevance, and theoretical foundation. We appreciate your supportive comments regarding the clarity of the objectives, the appropriateness of the methods, and the quality of the discussion and references. Your feedback is highly motivating and has reinforced our efforts to contribute meaningfully to the field of preventive health and self-management. Thank you again for your careful review.